# Simultaneous Determination of Seven Pesticides and Metabolite Residues in Litchi and Longan through High-Performance Liquid Chromatography-Tandem Mass Spectrometry with Modified QuEChERS

**DOI:** 10.3390/molecules27175737

**Published:** 2022-09-05

**Authors:** Siwei Wang, Xiaonan Wang, Qiang He, Haidan Lin, Hong Chang, Haibin Sun, Yanping Liu

**Affiliations:** 1Institute of Plant Protection, Guangdong Academy of Agricultural Sciences, Key Laboratory of Green Prevention and Control on Fruits and Vegetables in South China Ministry of Agriculture and Rural Affairs, Guangdong Provincial Key Laboratory of High Technology for Plant Protection, Guangzhou 510640, China; 2Guangdong Engineering Research Center for Insect Behavior Regulation, South China Agricultural University, Guangzhou 510642, China; 3Guangdong Quality Safety Center of Agricultural Products, Department of Agriculture and Rural Affairs of Guangzhou, Guangzhou 510230, China

**Keywords:** QuEChERS, HPLC-MS/MS, litchi and longan, pesticide and metabolite, residue

## Abstract

This study established a QuEChERS high-performance liquid chromatography/tandem triple-quadrupole mass spectrometry method for determining azoxystrobin, pyraclostrobin, picoxystrobin, difenoconazole, chlorantraniliprole, imidacloprid, and cyantraniliprole and its metabolite (IN-J9Z38) in litchi and longan, and applied this method to the real samples. The residues in samples were extracted with acetonitrile and purified with nano-ZrO_2_, C_18_, and PSA. The samples were then detected with multireactive ion monitoring and electrospray ionization in the positive ion mode and quantified using the external matrix-matched standard method. The results showed good linearities for the eight analytes in the range of 1–100 μg/L, with correlation coefficients (*r*^2^) of >0.99. The limit of quantification was 1–10 μg/kg, and the limit of detection was 0.3–3 μg/kg. Average recovery from litchi and longan was 81–99%, with the relative standard deviation of 3.5–8.4% at fortified concentrations of 1, 10, and 100 μg/kg. The developed method is simple, rapid, efficient, and sensitive. It allowed the rapid screening, monitoring, and confirming of the aforementioned seven pesticides and a metabolite in litchi and longan.

## 1. Introduction

Litchi and longan are both vital economic fruit trees in tropical and subtropical regions of China. Their annual output in China ranks highest in the world [1,2,3,4]. Litchi is known as the “king of fruits”, and longan is known as “southern ginseng” in China [5,6,7,8]. The high temperature and humidity in the planting areas of China make litchi and longan trees vulnerable to diseases and pests. At present, 140 pesticide products in litchi have been registered in China, including 48 active ingredients (10 insecticides, 23 fungicides, 1 herbicide, and 14 plant growth regulators) [9]. Seven pesticide products have been registered in longan, of which only 6 are active ingredients (2 insecticides, 1 fungicide, and 3 plant growth regulators) [9]. Most registered pesticides are pyrethroids, organophosphorus, triazole, and carbamate, and these chemical pesticides have been used for many years in litchi and longan orchards. The annual high-dose and frequent usage of pesticides has led to the prominent problems of resistance, pesticide residue pollution, and dose exceeding the standard in litchi and longan orchards [10]. The repeated use of chemical pesticides causes not only toxicity to fruits, but also environmental pollution and harm to non-target biological entities. For example, the rate of acute oral and contact toxicity of pyrethroid and organophosphorus insecticides to bees, fish, and birds is high [11,12,13]. An increasing number of studies have proven that fungicides could also harm the health of non-target organisms such as bees [14,15,16]. Triazole fungicides are classified as “potential human carcinogens” by the US Environmental Protection Agency [17,18]. Consumers are also increasingly concerned about the health and pollution problems caused by pesticides and their metabolite residues. The toxicity of some pesticide metabolites may even be higher than that of the parent pesticide [19]. The fruit quality and safety and environmental problems caused by pesticides deserve attention. Hence, establishing a method for the simultaneous determination of pesticides and their metabolites in litchi and longan is valuable.

At present, the main detection and analytical methods for the aforementioned pesticides are gas chromatography-electron capture detection (GC-ECD), gas chromatography-mass spectrometry (GC-MS), high-performance liquid chromatography (HPLC), and HPLC-mass spectrometry (HPLC-MS/MS) [20,21,22,23,24,25,26,27,28,29,30,31,32]. The main matrix includes vegetables such as broccoli and zucchini, fruits such as pears and apples, and traditional Chinese medicine [20,21,22,23,24,25,26,27,28,29,30,31,32]. Few studies have reported on multiresidue analysis of pesticides and their metabolites in litchi and longan. Common sample purification methods include solid phase extraction (SPE), liquid–liquid distribution extraction, and QuEChERS [20,21,22,23,24,25,26,27,28,29,30,31,32]. SPE is a tedious and time-consuming method involving the use of a large amount of organic solvents and nitrogen blowing, which is relatively cumbersome. Many types of commercial SPE columns are available, but selecting suitable SPE column for multiresidue analysis is difficult. The dispersive liquid–liquid extraction method requires a large amount of organic solvents and is prone to emulsification, which affects the recovery of target analytes. QuEChERS can achieve better purification with the addition of a small amount of adsorbent to the extraction solution to adsorb impurities. The operation of QuEChERS, which is currently a common purification method for fruits and vegetables, is fast and simple [33,34,35,36].

The main objectives of this paper are to establish and validate a fast and sensitive QuEChERS method employing a combination of nano-ZrO_2_, C_18_, and PSA for sample pretreatment before HPLC-MS/MS analysis for the simultaneous quantification of 7 pesticides and 1 metabolite in litchi and longan.

## 2. Materials and Methods

### 2.1. Materials and Reagents

The standards of azoxystrobin (99.5% purity), difenoconazole (99.0% purity), chlorantraniliprole (97.8% purity), pyraclostrobin (99.5% purity), and imidacloprid (98.5% purity) were purchased from Dr Ehrenstorfer GmbH (Wesel, Germany). The standard of picoxystrobin (98.4% purity) was purchased from Chem Service (West Chester, PA, USA). The standards of cyantraniliprole (98.0% purity) and its metabolite IN-J9Z38 (97.8% purity) were purchased from Toronto Research Chemicals (Toronto, Canada) and Hangzhou Trylead Chemical Technology Co., Ltd. (Hangzhou, China). HPLC grade acetonitrile (MeCN) and methanol were obtained from Fisher (Pittsburgh, PA, USA). Chromatographically pure formic acid was obtained from Fluka (Seeize, Germany). Analytical grade anhydrous magnesium sulfate (anhydrous MgSO_4_) and sodium chloride (NaCl) were obtained from Sinopharm Chemical Reagents Co., Ltd. (Shanghai, China). The octadecylsilane (C_18_, 40 μm) adsorbent was purchased from Agela Technologies Inc. (Tianjin, China). The graphitized carbon black (GCB, 120–400 MESH) and primary secondary amine (PSA, 40–60 μm) were obtained from ANPEL Laboratory Technologies Inc. (Shanghai, China). Nano-ZrO_2_ (99.99%, ≤100 nm) was from Aladdin Biochemical Technology Co., Ltd. (Shanghai, China), and multi-walled carbon nanotubes (MWCNTs; 95%, 10–30 μm, 8 nm) were from Nanjing XFNANO Materials Technology Co., Ltd. Moreover, the ultra-pure water was prepared using a Milli-Q Integral Water Purification System (Millipore Corporation; Burlington, MA, USA). The filter membrane (13 mm × 0.22 μm) was obtained from Ameritech Science and Technology Ltd. (Chicago, IL, USA). The high-speed refrigeration centrifuge model (GTR22-1) was also used (Beili Medicine Centrifuge Factory; Beijing, China).

### 2.2. Sample Pretreatment

The mashed litchi or longan sample (10.0 g) was weighed and extracted using 10 mL MeCN. The mixture was shaken for 2 min. Subsequently, 2 g NaCl and 4 g anhydrous MgSO_4_ were added. After mixing for an additional 1 min, the mixture was placed on a horizontal oscillator for 10 min. Samples were centrifuged for 5 min at 5204× *g*. Next, 2 mL of the supernatant was transferred into a 5 mL centrifuge tube containing 300 mg anhydrous MgSO_4_, 25 mg C_18_, 25 mg PSA, and 10 mg nano-ZrO_2_ for cleanup. After shaking for 1 min, samples were again centrifuged for 5 min at 5204× *g*, with supernatants then being passed through a 0.22 μm nylon syringe filter and transferred to an autosampler vial to conduct HPLC–MS/MS analyses.

### 2.3. Instrumental Parameters

A Shimadzu LC-20A HPLC system was used for separating the target analytes on a InfinityLab Poroshell 120 SB-C18 column (Dim:75 mm × 2.1 mm, 2.7 µm particle size, Agilent, Palo Alto, USA) by using a column oven at 35 °C. The separation was performed through gradient elution with A (0.1% of formic acid aqueous solution) and B (MeCN) as the mobile phase, and the flow rate was kept constant (0.3 mL/min) during the complete analysis process. The gradient program was: 0–2 min 80% A–5% A, 2–3.5 min 5% A, 3.5–4.5 min 20% A, 4.5–6 min 20% A. Injection (5 μL) was conducted using an autosampler.

A triple quadrupole mass spectrometer (Shimadzu 8045; Shimadzu; Kyoto, Japan) equipped with electrospray ionization in the positive ion mode (ESI^+^) was used to quantify the target analytes. The oven temperature was set at 350 °C, the desolvation line was set at 250 °C, the temperature of the heating block was 400 °C, nitrogen was used as a nebulizer and collision gas, and multireactive ion monitoring (MRM) was selected to analyze the target analytes with a dwell time of 80 ms. The optimal precursor ions, product ions, collision energies, and other instrument parameters for each analyte were acquired by directly infusing each target pesticide at a concentration of 100 μg/L standard solution into the ion source in the instrument. All other relevant MS parameters are listed in Table 1.

### 2.4. Method Validation

Recovery rate, linearity, and limit of quantification (LOQ) values were used to assess the accuracy and reliability of the developed techniques. The standards produced a linear result between 1 and 100 μg/L. Fortified blank food samples at concentrations of 1, 10, and 100 μg/kg were used to evaluate the recovery rate, and each spiked level was replicated six times, whereas the corresponding relative standard deviations (RSDs) represented the method precision. Based on the guidelines of SANTE/11813/2017, the LOQ for pesticide was considered as the lowest spiked level in the matrix.

## 3. Results and Discussion

### 3.1. Optimization of Chromatographic Separating Column

The separation effects of the two liquid chromatographic columns on the target analytes were compared. A: Shimadzu Shim-pack GIST-HP C18 (Dim: 50 mm × 2.1 mm, 3.0 µm particle size), and B: InfinityLab Poroshell C18 (Dim: 75 mm × 2.1 mm, 2.7 µm particle size). Chromatographic column A was a short column which has the advantage of a short analysis time, but the peak symmetry of the 8 target analytes was poor. Chromatographic column B exhibits good response values and peak symmetry. Hence, InfinityLab Poroshell C18 was considered to be the most desirable chromatographic column. The relevant chromatographs are shown in Figure 1.

### 3.2. Optimization of Mobile Phase

The mobile phase can influence the resolution effect, response value, and retention time of the target analytes. A shorter retention time is desirable if baseline separation can be achieved. Using the 75-mm InfinityLab Poroshell C18 chromatographic column, the effects of mobile phases A (methanol and 0.1% formic acid aqueous solution), B (MeCN and 0.1% formic acid aqueous solution), C (methanol and water), and D (MeCN and water) on the resolution and response intensity of the 8 target analytes were investigated. The results showed that the mobile phases C and D without formic acid had a great effect on the resolution, which led to asymmetric peak and peak broadened. The addition of ammonium formate, and formic acid to the aqueous phase is a common means to improve the chromatographic peak shape, the instrument response value, and ionization efficiency. Generally, the use of an acidic mobile phase is conducive to mass spectrometry detection in positive ion mode and formic acid is one of the most commonly used modifiers. With the addition of 0.1% formic acid to the aqueous phase, the mobile phase components of A and B promoted the formation of [M + H]^+^ ion peaks and improved the analytical sensitivity of the target analyte. The response of B to the target analyte was relatively higher. Hence, B (MeCN and 0.1% formic acid aqueous solution) was chosen as the mobile phase. The relevant chromatographs are shown in Figure 2.

### 3.3. Comparison of Constant and Gradient Elution

MeCN and 0.1% formic acid aqueous solution were selected as the mobile phase. The effects of constant elution (A) and gradient elution (B) on the resolution and response intensity of the target compounds were investigated. The peaks of the 8 target analytes were symmetrical, and no peak broadening phenomenon was observed at the gradient elution mode. The resolution of each target compound and the accuracy of quantitative results could be good. The relevant chromatographs are shown in Figure 3.

### 3.4. Optimization of MS/MS

The eight target analytes contained strong negative dielectrics such as oxygen and nitrogen atoms, which could form adduct ions ([M + H]^+^) with the hydrogen ions in the spray droplets under the positive ion mode (ESI^+^). The stability [M + H]^+^ of each pesticide was obtained using MS positive ion detection of 100 μg/kg of standard solution. To improve the ionization efficiency of the target analyte, the relevant parameters in the source were optimized. In the positive ion mode, the scanning range was set according to the relative molecular weight of the target compound, Q3 full scan on the compound was carried out, and stable [M + H]^+^ molecular ions were obtained through primary mass spectrometry scanning, and the parent ions were determined. Then, the target compound was scanned to obtain product ions. The CE, Q1 pre, Q3 pre, and product ions of the 8 target compounds were optimized in the MRM mode. Finally, the mass spectrum conditions of the target compounds were determined.

### 3.5. Optimization of Extraction Solvents

According to the relevant literature, the most commonly used extraction solvents for the QuEChERS method are ethyl acetate, acetone, and acetonitrile. Acetone cannot be separated from water in the absence of a non-polar solvent, but it can easily extract the pigment and other impurities from the matrix. In particular, the litchi pericarp contains more pigments, flavonoids, polyphenols, and other impurities, and the co-extraction phenomenon is more severe. The extraction efficiency of ethyl acetate is low when extracting polar pesticides, and ethyl acetate can be easily emulsified. Acetonitrile can be easily separated from water by addition of NaCl, and it extracts fewer impurities. Acetonitrile is the most commonly used extraction solvent in the QuEChERS method [28]. Hence, acetonitrile was chosen as the extraction solvent.

### 3.6. Optimization of Purification Adsorbents

Sample purification is a very important step that could avoid any potential contamination of the chromatographic column and MS detection. The QuEChERS method has the advantage of less organic solvent consumption, good reproducibility, and high sensitivity, that is widely applied for multi-residue analysis in fruits and vegetables. Hence, selecting appropriate adsorbents is of great importance for quantitative accuracy.

The adsorption effect and recovery rate of 8 target compounds with the five adsorbents (MWCNTs, nano-ZrO_2_, PSA, C_18_, and GCB) were investigated. The MWCNTs belongs a new conductive carbon nanometer material with a larger specific surface area, and with a good purification effect on pigments. In this study, the recoveries of the 8 target compounds were low (ranged from 1% to 56% except picoxystrobin (88%) and imidacloprid (77%)) when the dosage of MWCNTs was 5–10 mg, indicating that the MWCNTs adsorbent has a strong adsorption capacity for the target compound and is not easy to desorb. Although GCB could significantly remove the pigment from the sample, the recovery rate was low due to the strong adsorption of GCB on difenoconazole, chlorantraniliprole, cyantraniliprole, and IN-J9Z38. The nano-ZrO_2_, PSA, and C18 could satisfy the requirement of a recovery range of 70−120%, exhibited good recoveries of the target compounds (90–103%, Figure 4).

### 3.7. Optimization of Adsorbent Dosage

The purification effect of a single adsorbent for co-extracted impurities is considerably weaker than that of combined adsorbents, especially for litchi and longan, which are relatively complex matrices. Therefore, the combination of different adsorbents was tested to establish the most effective approach to producing a purified sample with satisfactory recovery. The advantages of C_18_ adsorbent are its large specific surface area, which can effectively remove non-polar interferences such as fats and lipids. PSA is a weak anion exchange filler that can effectively remove fatty acids, pigments, sugars, and other substances from the matrix. Nano-ZrO_2_ has small particle size, large specific surface area, and removes lipophilic impurities. The recovery of the target compounds ranged from 86% to 103% when the dosage of PSA, C_18,_ and nano-ZrO_2_ was in the range of 25–50 mg, 25–50 mg, and 10–20 mg (Table 2). As such, a combination of 25 mg C_18_ + 25 mg PSA + 10 mg nano-ZrO_2_ was selected as the adsorbent of choice for these analyses.

### 3.8. Matrix Effect

The ME refers to the influence of components other than analytes in the sample on the response value of the analytes [33]. ME is an important factor affecting the accuracy of HPLC-MS/MS quantitative results [37]. ME (%) = [(m_matrix_/m_solvent_) − 1)] × 100%, where m_matrix_ is the slope of the matrix matching standard curve, and m_solvent_ is the slope of the pure solvent standard curve. A 100 mg/L mixed standard solution was diluted with litchi and longan matrix purification solutions stepwise to prepare the matrix standard curve. A positive ME indicates the matrix enhancement effect, and the matrix can improve the response of the target. A negative ME indicates the matrix inhibition effect, and the matrix can reduce the response of the target [34]. The ME is divided into 3 grades according to the absolute value of ME. When the absolute ME value is 0–20%, the ME is weak; when the absolute ME value is 20–50%, the ME is medium; and when the absolute ME value is >50%, the ME is strong [38,39,40,41]. Table 3 shows that only the ME of pyraclostrobin was in the range of 0–20% in longan, indicating the presence of a weak ME. Azoxystrobin, picoxystrobin, difenoconazole, chlorantraniliprole, cyantraniliprole, and IN-J9Z38 in longan have a medium ME (0.5–50%). In this study, the matrix matching standard solution was used to correct the ME.

### 3.9. Method Validation

The recovery rates, RSDs, limit of detection (LODs), and LOQs of the 7 pesticides and a metabolite residue in litchi and longan are shown in Table 4. Good linearity was acquired for correlation coefficient values of >0.99. The average recovery rate was 81–99% in matrices, with an RSD of 3.5–8.4%, which represent satisfactory precision and accuracy. These results were compliant with the rules stating that the mean recovery rate should be in the range of 70–120% with an associated RSD of ≤20%. The LOQ was 1–10 μg/kg for both litchi and longan. The LOD was the concentration that produced a signal-to-noise ratio of 3 and was 0.3–3 μg/kg.

### 3.10. Application in Real Samples

Ten litchi and longan samples randomly purchased from the market were examined using the validated HPLC-MS/MS method for monitoring the aforementioned 8 target analytes. Azoxystrobin, pyraclostrobin, difenoconazole, and chlorantraniliprole were detected. Among them, the MRL value of azoxystrobin, pyraclostrobin, and difenoconazole were established in litchi [42]. The residues in the real samples did not exceed the regulated MRLs (Table 5).

## 4. Conclusions

We simultaneously determined and analyzed azoxystrobin, pyraclostrobin, picoxystrobin, difenoconazole, chlorantraniliprole, imidacloprid, and cyantraniliprole, and a cyantraniliprole metabolite IN-J9Z38 in litchi and longan by using optimized sample pretreatment, instrument conditions, and QuEChERS–HPLC-MS/MS. The samples were homogeneously extracted with acetonitrile, purified using the improved QuEChERS method, and detected through HPLC-MS/MS. The Infinity Lab Poroshell 120 SB-C18 chromatographic column, electrospray ionization, positive ion scanning, and MRM were used for separation, analysis, detection, and quantification of the target analytes. The matrix matching standard solution was determined using the external standard method. The linear relationship of the 8 target compounds within the range of 1–100 μg/L was good, and the correlation coefficients were >0.99. At the spiked level of 1, 10, and 100 μg/kg, the average recovery of eight target compounds was 81–99% and the RSD was 3.5–8.4%. The pretreatment process of the method is simple and rapid, and the detection limit, precision, and linear range of the method could meet the requirements. This method can be used for the simultaneous determination and analysis of the aforementioned pesticide residues in litchi and longan samples.

## Figures and Tables

**Figure 1 molecules-27-05737-f001:**
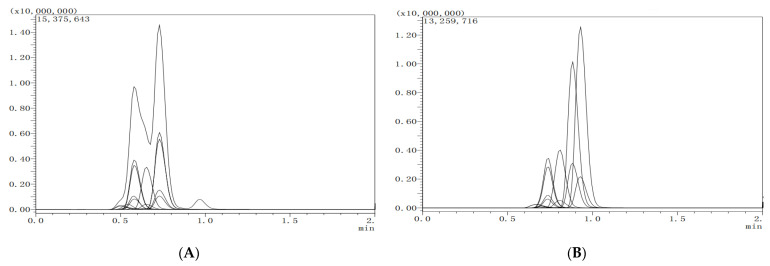
The total ion chromatogram of eight targeted compounds separated by the two different columns (lateral axis is time, and longitudinal axis is response intensity). (**A**) Shim-pack GIST-HP C18, 50 mm × 2.1 mm, 3.0 μm; (**B**) InfinityLab Poroshell C18, 75 mm × 2.1 mm, 2.7 μm.

**Figure 2 molecules-27-05737-f002:**
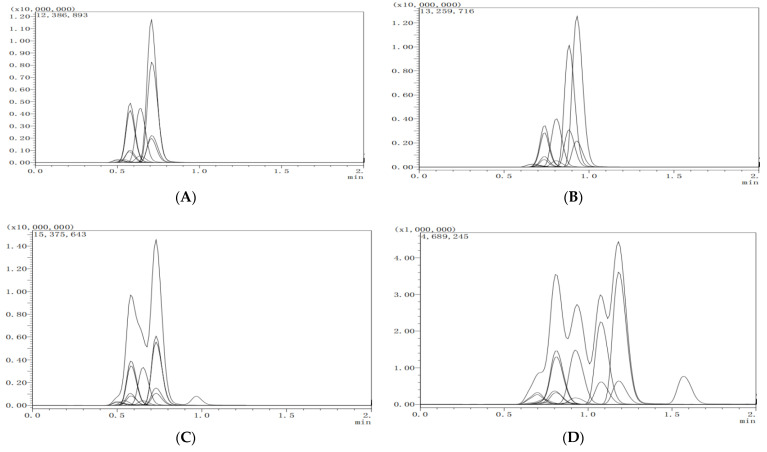
The chromatogram of eight targeted compounds separated by the different mobile phase components (lateral axis is time, and longitudinal axis is response intensity). (**A**) Methanol-water containing 0.1% formic acid; (**B**) acetonitrile-water containing 0.1% formic acid; (**C**) methanol-water; (**D**) acetonitrile-water.

**Figure 3 molecules-27-05737-f003:**
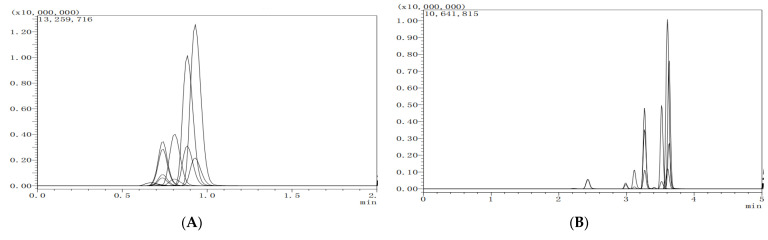
The chromatogram of eight targeted compounds separated by the constant elution (**A**) and gradient elution, (**B**) of mobile phase (lateral axis is time, and longitudinal axis is response intensity).

**Figure 4 molecules-27-05737-f004:**
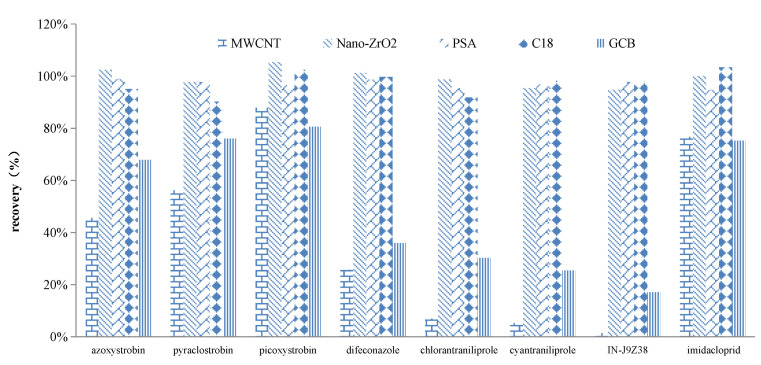
Absorption efficiency of five different adsorbent for eight targeted compounds.

**Table 1 molecules-27-05737-t001:** Mass parameters of eight targeted compounds.

Compound	Precursor Ion (*m*/*z*)	Daughter Ion (*m*/*z*)	CE/eV	Q1Pre (V)	Q3Pre (V)	Retention Time RT/min
Azoxystrobin	404.20	344.10 *, 329.10	−25, −30	−15, −19	−13, −22	3.25
Pyraclostrobin	388.15	163.10 *, 133.05	−24, −36	−29, −14	−17, −24	3.62
Picoxystrobin	368.00	145.10 *, 117.25	−23, −38	−28, −29	−27, −22	3.51
Difenoconazole	406.10	251.00 *, 337.00	−26, −18	−12, −19	−30, −24	3.59
Chlorantraniliprole	484.00	285.90 *, 453.35	−12, −14	−23, −18	−29, −23	2.97
Cyantraniliprole	475.31	285.95 *, 444.10	−10, −19	−30, −24	−15, −18	3.33
IN-J9Z38	457.10	299.00 *, 188.00	−27, −35	−20, −17	−26, −14	3.42
Imidacloprid	256.10	209.05 *, 175.10	−14, −18	−18, −19	−22, −18	2.40

* Quantitative ion.

**Table 2 molecules-27-05737-t002:** Effect of adsorbent dosage on the recovery of eight targeted compounds in litchi and longan samples (*n* = 5).

Different Adsorbent Combination	Recovery/%							
	Azoxystrobin	Pyraclostrobin	Picoxystrobin	Difenoconazole	Chlorantraniliprole	Cyantraniliprole	IN-J9Z38	Imidacloprid
25 mg PSA + 25 mg C_18_	100 ± 2.5	100 ± 2.0	100 ± 3.5	98 ± 4.0	100 ± 2.0	99 ± 2.7	98 ± 5.1	100 ± 2.1
25 mg PSA + 10 mg nano-ZrO_2_	96 ± 3.2	96 ± 4.0	99 ± 2.5	102 ± 1.5	97 ± 3.5	98 ± 4.0	97 ± 4.0	99 ± 3.8
25 mg C_18_ + 10 mg nano-ZrO_2_	99 ± 3.8	102 ± 2.0	100 ± 2.1	99 ± 3.5	96 ± 4.9	97 ± 3.2	91 ± 1.5	100 ± 2.7
50 mg PSA + 50 mg C_18_	97 ± 4.2	99 ± 1.0	102 ± 2.7	103 ± 3.1	102 ± 1.5	100 ± 1.5	100 ± 2.5	103 ± 2.5
50 mg PSA + 20 mg nano-ZrO_2_	88 ± 2.0	97 ± 2.5	93 ± 2.5	96 ± 3.5	94 ± 3.1	95 ± 3.1	90 ± 1.5	97 ± 3.6
50 mg C_18_ + 20 mg nano-ZrO_2_	89 ± 1.0	95 ± 5.0	92 ± 4.4	93 ± 3.8	94 ± 3.0	91 ± 6.8	90 ± 4.0	99 ± 3.0
25 mg PSA + 25 mg C_18_ + 10 mg nano-ZrO_2_	90 ± 2.1	91 ± 2.5	90 ± 3.5	90 ± 3.5	92 ± 2.5	90 ± 1.5	88 ± 2.0	94 ± 5.0
50 mg PSA + 50 mg C_18_ + 20 mg nano-ZrO_2_	87 ± 7.6	94 ± 3.6	88 ± 3.6	89 ± 2.3	90 ± 1.5	88 ± 5.1	86 ± 3.1	93 ± 4.5

**Table 3 molecules-27-05737-t003:** Linear equations and matrix effect of eight targeted compounds in litchi and longan samples (*n* = 5).

Compound	Sample	Linear Range /μg/L	Linear Equation	Correlation Coefficient /R^2^	ME ^a^ (%)
Azoxystrobin	solvent	1–100	Y = 1.3988 × 10^8^X + 114,089	0.9988	
	Litchi	1–100	Y = 5.73765 × 10^7^X – 11,998.3	0.9989	−59.03
	Longan	1–100	Y = 7.0 × 10^7^X + 900,000	0.998	−50.01
Pyraclostrobin	solvent	1–100	Y = 2.21447 × 10^8^C + 340,737	0.9906	
	Litchi	1–100	Y = 5.56824 × 10^7^X + 853,513	0.9996	−74.92
	Longan	1–100	Y = 2.22603 × 10^8^X – 499,085	0.9902	0.52
Picoxystrobin	solvent	1–100	Y = 1.40371 × 10^8^X – 129,785	0.9998	
	Litchi	1–100	Y = 2.59173 × 10^7^X − 7809.87	0.9998	−81.52
	Longan	1–100	Y = 7.20201 × 10^7^X – 97,424.8	0.997	−48.67
Difenoconazole	solvent	1–100	Y = 3.14863 × 10^8^X – 664,206	0.9978	
	Litchi	1–100	Y = 1.39059 × 10^8^X + 132,133	0.9992	−55.82
	longan	1–100	Y = 1.0 × 10^8^X + 2 × 10^7^	0.9901	−48.19
Chlorantraniliprole	solvent	1–100	Y = 3.48823 × 10^7^X + 43,168.4	0.9958	
	litchi	1–100	Y = 1.45219 × 10^7^X + 190,031	0.999	−58.41
	longan	1–100	Y = 2.00116 × 10^7^X + 14,557.3	0.9954	−42.6
Cyantraniliprole	solvent	1–100	Y = 7.59264 × 10^6^X − 4650.42	0.9998	
	litchi	1–100	Y = 2.38347 × 10^6^X + 685.684	0.9989	−65.42
	longan	1–100	Y = 3.98107 × 10^6^X − 3974.61	0.9962	−43.01
IN-J9Z38	solvent	1–100	Y = 2.41814 × 10^6^X − 3692.34	0.9972	
	litchi	1–100	Y = 3.61443 × 10^5^X + 908.734	0.992	−79.12
	longan	1–100	Y = 1.62265 × 10^6^X + 3504.66	0.9917	−33.51
Imidacloprid	solvent	1–100	Y = 2.19412 × 10^7^X + 11,740.08	0.9962	
	litchi	1–100	Y = 6.56189 × 10^6^X − 1297.32	0.9989	−70.14
	longan	1–100	Y = 5.0 × 10^6^X + 68,417	0.999	−77.19

^a^ ME means matrix effect, and was calculated with the equation: ME = (slope of the matrix-matched standard/slope of the solvent standard − 1) × 100%. An ME with a negative and positive value represents that the pesticide response is suppressed and enhanced. It is generally believed that when: |ME| < 20% the matrix does not exist; 20% ≤ |ME| ≤ 50% it indicates a medium matrix effect; and when |ME| > 50% it indicates a strong matrix effect.

**Table 4 molecules-27-05737-t004:** Recoveries, RSDs, LOD, and LOQ of eight targeted compounds in litchi and longan samples (*n* = 5).

Compound	Sample	Spiked Level /(μg/kg)	Average Recoveries ^a^ /%	Relative Deviation ^b^ /%	LOD /(μg/kg)	LOQ /(μg/kg)
Azoxystrobin	litchi	1, 10, 100	83, 94, 94	3.9, 7.5, 6.9	0.3	1
	longan	1, 10, 100	81, 89, 94	3.5, 6.8, 8.2	0.3	1
Pyraclostrobin	litchi	1, 10, 100	86, 94, 95	7.1, 8.2, 8.1	0.3	1
	longan	1, 10, 100	85, 96, 94	6.0, 7.5, 4.0	0.3	1
Picoxystrobin	litchi	1, 10, 100	84, 94, 95	4.4, 6.6, 7.4	0.3	1
	longan	1, 10, 100	86, 92, 96	7.1, 8.4, 4.7	0.3	1
Difenoconazole	litchi	1, 10, 100	84, 91, 94	4.1, 6.1, 5.0	0.3	1
	longan	1, 10, 100	90, 97, 96	5.0, 4.1 5.4	0.3	1
Chlorantraniliprole	litchi	1, 10, 100	85, 93, 93	5.8, 6.3, 4.7	0.3	1
	longan	1, 10, 100	82, 95, 94	4.3, 6.1, 7.0	0.3	1
Cyantraniliprole	litchi	1, 10, 100	85, 96, 99	4.7, 6.1, 4.7	0.3	1
	longan	1, 10, 100	84, 91, 99	5.2, 6.7, 5.6	0.3	1
IN-J9Z38	litchi	1, 10, 100	85, 89, 95	4.3, 5.6, 7.7	3	10
	longan	1, 10, 100	86, 96, 98	5.2, 7.5, 4.7	3	10
Imidacloprid	litchi	1, 10, 100	83, 97, 97	4.5, 7.0, 5.1	0.3	1
	longan	1, 10, 100	84, 95, 98	5.3, 6.3, 5.4	0.3	1

^a^ The recovery was calculated by the formula: Recovery = *C_d_ /C_s_* × 100%, where *C_d_* represents the detected concentration and *C_s_* represents the spiked concentration. Results were expressed as mean ± standard deviation (SD) with 95% confidence intervals. ^b^ Mean value of five determinations.

**Table 5 molecules-27-05737-t005:** Residues of eight targeted compounds in real litchi and longan samples.

Matrix MRLs	Azoxystrobin	Pyraclostrobin	Picoxystrobin	Difenoconazole	Chlorantraniliprole	Cyantraniliprole	IN-J9Z38	Imidacloprid
	mg/kg							
Litchi	ND-0.159	ND-0.091	ND	ND-0.458	ND-0.019	ND	ND	ND
Longan	ND-0.17	ND-0.08	ND	0.02–0.16	0.02–0.03	ND	ND	ND
MRLs in litchi	0.5	0.1	-	0.5	-	-	-	-
MRLs in longan	-	-	-	-	-	-	-	-

Note: ND is none of detection.

## Data Availability

Not applicable.

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
