# Peer review of "Simultaneous Determination of Seven Pesticides and Metabolite Residues in Litchi and Longan through High-Performance Liquid Chromatography-Tandem Mass Spectrometry with Modified QuEChERS"

_molecules, 2022, doi:10.3390/molecules27175737_

Round 1

Reviewer 1 Report (New Reviewer)

The authors in the submitted manuscript entitled “Simultaneous Determination of Seven Pesticides and Metabolite Residues in Litchi and Longan Through High-Performance 3 Liquid Chromatography-Tandem Mass Spectrometry with 4 Modified QuEChERS” analyzed seven pesticides in litchi and longan samples.

Overall the manuscript is interesting and fit the scope of the journal but before acceptance same improvements are necessary. It is contain some typing errors that need to be improved in the first place.

In abstract

Line 21 rewrite it (or delete some part). It is evident that chromatographic column was used…

Introduction part

Line 36 in the planting

Line 73 spaces are missing and simple [33-35]

Cite and analyse the following work:

https://doi.org/10.1016/j.foodchem.2016.06.114

Materials and reagent

Line 86 What does it mean: chromatographically pure HPLC-grade, gradient grade or LC-MS grade?

The title of table 1 is not correct. There are no spectra.

Figure with formulas could be helpful.

Line 132 linearity instead of linear equation

Results and Discussion

Line 142-143 Why these two columns were tried?

Line 164 modifier instead of reagents

Author Response

(1)Line 21 rewrite it (or delete some part). It is evident that chromatographic column was used… Response: Done. We have rewritten it. (2)Line 36 in the planting Response: Done. We have changed. (3)Line 73 spaces are missing and simple [33-35] Response: Done. We have changed. (4)Cite and analyse the following work: https://doi.org/10.1016/j.foodchem.2016.06.114 Response: Done. We have cited. (5)Line 86 What does it mean: chromatographically pure HPLC-grade, gradient grade or LC-MS grade? Response: Done. We have changed. (6) The title of table 1 is not correct. There are no spectra. Response: Done. We have changed. (7)Figure with formulas could be helpful. Response: Because there are eight pesticides, it will be crowded to put the molecular formula on Figures, which will affect the display effect. (8)Line 132 linearity instead of linear equation. Response: Done. We have changed. (9)Line 142-143 Why these two columns were tried? Response: These two columns were commonly used and had good separation effect for most pesticides. Therefore, we chose them for comparison. (10)Line 164 modifier instead of reagents Response: Done. We have changed.

Reviewer 2 Report (New Reviewer)

Authors have carried out useful interesting work and reported HPLC-MS method for quantifying several pesticides in litchi and longan.the lod and loq values are quite impressive. Results are well supported  with relevant statistical analysis. In my opinion article can be accepted in current form.

Author Response

None.

This manuscript is a resubmission of an earlier submission. The following is a list of the peer review reports and author responses from that submission.

Round 1

Reviewer 1 Report

The manuscript presents in a clear and didactic way the steps involved (sample preparation, HPLC conditions, ionization and detection conditions , matrix effect) during the development and validation of an analytical method for the determination of 7 pesticides and 1 metabolite in litchi and longan through high-performance liquid chromatography-tandem mass spectrometry. Despite the QuEChERS method being widely used, the authors explored the use of different adsorbents and their combined use for better extraction/purification of the analytes present in the sample and obtained satisfactory results in the analysis of the fruits under study, applying the validated method for sample analysis real.

I believe that this manuscript will be an excellent reference material for the analytical area. As a suggestion, all that remained was to discuss/justify the reason for not using an internal standard in the analysis. I also recommend citing the classical article by Sapozhnikova & Lehotay (2013) on the matrix effect.

Sapozhnikova Y, Lehotay SJ. 2013. Multi-class, multi-residue analysis of pesticides, polychlorinated biphenyls, polycyclic aromatic hydrocarbons, polybrominated diphenyl ethers and novel flame retardants in fish using fast, low-pressure gas chromatography–tandem mass spectrometry. Anal Chim Acta. 758:80-92 doi:10.1016/j.aca.2012.10.034.

Author Response

  1. We used matrix matched standard in the method to avoid the matrix effect and correcting extraction recoveries, and discussed it in the manuscript .
  2.  We have cited the classical article by Sapozhnikova & Lehotay (2013) on the matrix effect.

Reviewer 2 Report

The manuscript submitted by Wang and co-authors illustrates a new method for the determination of residual pesticides detected on litchi and longan, based on HPLC-MS/MS.

The meaning of reported results is limited by the experimental design; the standards have not been analysed in mixture. The reported data was separately acquired for each targeted compound. Or, during ESI experiments, ion suppression phenomena might be noted.

The authors tried the mobile phase free of any salt, or they chose positive MS detection. How the ionization can be done? This part should be considered.

The quality of the data presentation is low and should completely be considered before re-submission. Here are a few examples:

All the figures have low resolution, and the different curves are indistinguishable as represented in grey scale.

Moreover, the labels of the axes (chromatograms) are missing everywhere.

Figure 4 – the data does not correspond to any MS spectra as mentioned in the caption.

For this figure one or two examples would be enough, the remaining should be included in the Supplementary Information Part.

Table 1 – the m/z values are given with different number of decimals; last column is partially written in Chinese characters…

Author Response

Please let me express our sincere gratitude to your suggestions on revising our manuscript. We carefully checked the whole manuscript and seriously considered the comments. We have revised it according to the suggestions, and resubmitted it to your journal.

  1. The mixed standards were adopted in the experiment (including optimization of instrument and method). All shown in Fig.1, Fig.2, and Fig.3 were total ion chromatography (chromatography of mixed standards).
  2. The main purpose of adding salt in the mobile phase is to improve the peak shape. After the salt is added, it needs to be washed with a lot of water and then with 95% acetonitrile, otherwise it will be retain in the chromatographic column leading to weak resolution ability. 

    Ion source is a device that makes compounds produce ions in LC-MS/MS, and ESI is commonly used ion source. Controlling the fragmentation of ions by adjusting the voltage of ion source.

  3. We have changed all clear chromatograms according to suggests. 

Round 2

Reviewer 2 Report

The manuscript has not been improved and therefore, cannot be accepted for publication. 

Author Response

The combined use of QuEChERS and HPLC-MS/MS has been applied for the first time to the multiresidue detection of 7 pesticides and  metabolites in litchi and longan. It is of great importance to develop a sensitive, accurate and robust simultaneous analytical method for evaluation of the risks posed by these pollutants to human health which provides guidance to laboratory for the validation of methods for pesticide residues analysis in foodstuff.

We consider that this is a remarkable work that has to offer to the community and is worth publishing.